# Selective Detection of Nitrogen-Containing Compound Gases

**DOI:** 10.3390/s19163565

**Published:** 2019-08-15

**Authors:** Ran Yoo, Hyun-Sook Lee, Wonkyung Kim, Yunji Park, Aran Koo, Sang-Hyun Jin, Thang Viet Pham, Myung Jong Kim, Sunglyul Maeng, Wooyoung Lee

**Affiliations:** 1Department of Materials Science and Engineering, Yonsei University, 50 Yonsei-ro, Seodaemun-gu, Seoul 03722, Korea; 2School of Nano & Materials Science and Engineering, Kyungpook National University, 2559 Gyeongsang-daero, Gyeongsangbuk-do 37224, Korea; 3Isenlab Inc., Halla Sigma Valley, Dunchon-daero 545, Jungwon-gu, Seongnam-si, Gyeonggi-do 13215, Korea; 4Department of Electrical and Electronic Engineering, Woosuk University, 443, Samnye-ro, Samnye-eup, Wanju_Gun, Jeollabuk-do 55338, Korea; 5Functional Composite Materials Research Center and Division of Nano & Information Technology of KIST School, Korea Institute of Science and Technology, Jeonbuk 565-905, Korea

**Keywords:** chemical sensors, nitrogen-containing compound gases, trimethylamine, triethylamine, ammonia, NO, NO_2_, Al-doped ZnO nanoparticles, WO_3_ thin film, N-doped graphene

## Abstract

N-containing gaseous compounds, such as trimethylamine (TMA), triethylamine (TEA), ammonia (NH_3_), nitrogen monoxide (NO), and nitrogen dioxide (NO_2_) exude irritating odors and are harmful to the human respiratory system at high concentrations. In this study, we investigated the sensing responses of five sensor materials—Al-doped ZnO (AZO) nanoparticles (NPs), Pt-loaded AZO NPs, a Pt-loaded WO_3_ (Pt-WO_3_) thin film, an Au-loaded WO_3_ (Au-WO_3_) thin film, and N-doped graphene—to the five aforementioned gases at a concentration of 10 parts per million (ppm). The ZnO- and WO_3_-based materials exhibited n-type semiconducting behavior, and their responses to tertiary amines were significantly higher than those of nitric oxides. The N-doped graphene exhibited p-type semiconducting behavior and responded only to nitric oxides. The Au- and Pt-WO_3_ thin films exhibited extremely high responses of approximately 100,000 for 10 ppm of triethylamine (TEA) and approximately −2700 for 10 ppm of NO_2_, respectively. These sensing responses are superior to those of previously reported sensors based on semiconducting metal oxides. On the basis of the sensing response results, we drew radar plots, which indicated that selective pattern recognition could be achieved by using the five sensing materials together. Thus, we demonstrated the possibility to distinguish each type of gas by applying the patterns to recognition techniques.

## 1. Introduction

Many N-containing gases exude irritating odors, such as ammonia (NH_3_), trimethylamine (TMA), triethylamine (TEA), nitric oxide (NO), and nitrogen dioxide (NO_2_). NH_3_ mainly arises from natural sources through the decomposition of organic matter containing nitrogen. Exposure to high levels of NH_3_ emitted from chemical plants, cultivated farmland (fertilizer), and motor vehicles can cause irritation and serious burns on the skin and in the mouth, throat, lungs, and eyes [1,2].

TMA is a colorless, hygroscopic, and flammable tertiary amine that has a strong fishy odor at low concentrations and an NH_3_-like odor at higher concentrations. Exposure to high levels of TMA can cause headaches, nausea, and irritation to the eyes and respiratory system. After marine fish death, bacterial or enzymatic actions rapidly convert trimethylamine oxide into TMA—a volatile base that is largely responsible for the characteristic odor of dead fish [3,4]. Accordingly, the detection of TMA is essential for evaluating the freshness of fish [5,6,7]. TEA is a colorless volatile liquid with a strong fishy odor, reminiscent of the smells of NH_3_ and the hawthorn plant [8]. It is commonly utilized as a catalyst and an acid neutralizer for condensation reactions, and is useful as an intermediate for manufacturing medicines, pesticides, and other chemicals. It is also a decomposition product of the V-series nerve gas agent [9]. Short-term exposure to TEA can irritate the skin and mucous membranes of humans. Chronic (long-term) exposure of workers to TEA vapor can cause reversible corneal edema [10]. 

NO is a nonflammable, extremely toxic, oxidizing gas with a sharp sweet odor. NO can be released by the reaction of nitric acid with metals, e.g., in metal etching and pickling, and is a byproduct of the combustion of substances in fossil fuel plants and automobiles. NO is a skin, eye, and mucous membrane irritant, as moisture and O_2_ convert nitric oxide into nitric and nitrous acids. The most hazardous effects of NO are on the lungs. Inhalation causes symptoms such as coughing and shortness of breath, along with a burning sensation in the throat and chest [11]. NO is spontaneously converted to NO_2_ in air; thus, some NO_2_ is likely to be present when nitric oxide is detected in air [12]. NO_2_ has a strong harsh odor, similar to chlorine, and may exhibit a vivid orange color. The major source of NO_2_ is the burning of coal, oil, and gas. Almost all NO_2_ comes from motor-vehicle exhaust, metal refining, electricity generation from coal-fired power plants, and other manufacturing industries [13]. The reaction of NO_2_ with chemicals produced by sunlight leads to the formation of nitric acid, which is a major constituent of acid rain [14]. NO_2_ also reacts with sunlight, which leads to the formation of ozone and smog in air [15,16]. The main effect of breathing high levels of NO_2_ is an increased risk of respiratory problems, such as asthma, wheezing, coughing, colds, the flu, and bronchitis [17,18]. The U.S. National Institute for Occupational Safety and Health (NIOSH) has established exposure limits for these gases, as shown in Table 1 [19].

Various sensing materials have been investigated for the detection of N-containing compound gases. For the detection of tertiary amines (TMA, TEA), metal oxides, such as TiO_2_, WO_3_, MoO_3_, LaFeO_3_, SnO_2_, and ZnO, have been tested [20,21,22,23,24,25,26,27,28,29,30,31,32]. Sensing materials based on WO_3_, MoSe_2_, multi-walled C nanotubes, and graphene oxide have been reported to exhibit good sensitivity to NH_3_ gas [33,34,35,36,37,38,39,40,41]. Nitric oxide (NO, NO_2_) sensing has recently been performed using metal oxides, such as ZnO, SnO_2_, and WO_3_, and metal–polymer composites, such as nickel phthalocyanine (NiPc) and graphene [42,43,44,45,46,47,48,49,50,51,52,53]. However, the highly sensitive and selective detection of N-containing compounds is urgently required.

In this study, we investigated the sensing properties of Al-doped ZnO (AZO) nanoparticles (NPs), Pt-loaded AZO (Pt-AZO) NPs, a Pt-loaded WO_3_ (Pt-WO_3_) thin film, a Au-loaded WO_3_ (Au-WO_3_) thin film, and N-doped graphene toward NH_3_, TMA, TEA, NO, and NO_2_. We found that each N-based hazardous gas reacted distinctively to the five types of sensing materials, producing different sensing patterns.

## 2. Materials and Methods

### 2.1. Synthesis

#### 2.1.1. AZO NPs

AZO NPs were synthesized via a hydrothermal method [54,55]. Zinc acetate dehydrate (Zn(AC)_2_·2H_2_O, 99%, Sigma-Aldrich, Seoul, Korea) and potassium hydroxide (KOH, 99%, Sigma-Aldrich) were dissolved in methanol with a molar ratio of 1:3. Aluminum acetate (99%, Sigma-Aldrich) was placed into the zinc acetate solution to achieve 1.0 at% of doped Al. The KOH solution was mixed with the zinc acetate solution via stirring at 60 °C for 24 h. Then, the suspension was centrifuged and washed with methanol three times. The obtained samples were dried at 90 °C for 60 min and annealed at 350 °C for 30 min in a H_2_/N_2_ atmosphere.

#### 2.1.2. Pt-AZO NPs

For synthesizing Pt-AZO NPs, Pt NPs were coated on the surface of the as-synthesized AZO NPs with a deposition rate of 6−7 nm/min using a DC magnetron sputtering system in an agitated vessel [55]. In the agitated vessel, the powders were continuously stirred using a rotating impeller, and the Pt NPs were homogenously loaded on the surface of the AZO NPs. The Pt-loaded samples were prepared with a deposition time of 2 min.

#### 2.1.3. Pt-WO_3_ and Au-WO_3_ Thin Films

For synthesizing Pt-WO_3_ and Au-WO_3_ thin films, WO_3_ thin films were prepared via dual ion beam sputtering [56]. A tungsten metal target of 99.99% purity was employed. The WO_3_ was deposited onto an interdigitated Pt electrode formed on a Si/SiO_2_ wafer via a photolithography process. The dual ion beam consisted of a primary ion beam applied to the target and a secondary ion beam with accelerated atoms to be deposited on the substrate. The tungsten target was sputtered under the following conditions: The power of the main ion gun was 90 W, the voltage of the anode was 50 V, and the voltage of the cathode was −50 V. O ions were applied under the following conditions: The power of the assistant ion gun was 120 W, the voltage of the anode was 1000 V, and the voltage of the cathode was 300 V. The thickness of the WO_3_ thin film was 200 nm. Pt (2 nm) and Au (2 nm) were deposited on the WO_3_ thin film via direct current (DC) magnetron sputtering as catalysts. The thickness of Au and Pt was adjusted to ~2 nm by controlling the deposition time, where the deposition rate was 0.67 nm/s for Au and 0.28 nm/s for Pt. The samples were heat-treated at 550 °C for 1 h.

#### 2.1.4. N-Doped Graphene

N-doped graphene was synthesized via arc discharging. A hollow graphite rod with a size of 6 mm, bismuth oxide as a catalyst, and 4-aminobenzoic acid as a dopant were placed into the hole and discharged while inducing 150 A in a 550 Torr H_2_/He atmosphere used as a buffer. The amount of N in the graphene was 2 wt% [57].

#### 2.1.5. Summary of Sensing Materials

Table 2 presents the fabrication methods and specifications for the five aforementioned sensing materials used in the experiments.

### 2.2. Characterization

The morphology and shape of the as-synthesized sensing materials were investigated via field-emission scanning electron microscopy (FE-SEM, JEOL 7001F) and transmission electron microscopy (TEM, JEOL JEM-ARM200F). 

### 2.3. Device Fabrication

For fabrication of the gas sensor, interdigitated Cr (20 nm) and Pt (100 nm) electrodes were deposited on the patterned SiO_2_ substrate via DC magnetron sputtering [54,55]. The synthesized NPs (AZO and Pt-AZO NPs) were mixed with an α-terpineol binder and coated onto the interdigitated electrodes. The sensor was heat-treated at 300 °C for 1 h to remove the binder and annealed at 600 °C for 1 h. The Pt-WO_3_ and Au-WO_3_ thin films were sputtered directly onto the interdigitated electrodes. N-doped graphene was drop-coated onto the electrodes.

### 2.4. Gas Sensing Measurement

A device was mounted in a chamber of a tube furnace system and placed in a flow system equipped with gas cylinders and mass flow controllers (MFCs) to perform the gas sensing test. The working temperature of the sensor was controlled using the temperature controller of the tube furnace. With the application of controlled heat, the resistance of the sensing material was measured in the presence of synthetic air and then in the presence of air with a controlled amount of target gas. The amount of target gas was controlled to 10 parts per million (ppm) by varying the gas flow rates using the MFCs. All the gas sensing measurements were conducted at an operating temperature of 400 °C, except for N-doped graphene (room temperature). The sensing properties were measured using a combination of a current source (Keithley 6220) and a nanovoltmeter (Keithley 2182) with a constant current supply of 10 nA.

## 3. Results and Discussion

Figure 1 shows the size and morphology of the sensing materials. Figure 1a presents a TEM image of AZO NPs, which were spherical and had a diameter of ~25 nm. The isolated AZO NP represented the single crystallinity of a hexagonal wurtzite structure of ZnO with a lattice spacing of ~0.28 nm, which was confirmed by high-resolution TEM analysis with the electron diffraction pattern [54]. Figure 1b shows a TEM image of the as-synthesized Pt-AZO NPs. This indicates that the Pt NPs with a size of ~2 nm were uniformly distributed on the surface of the AZO NPs, which was confirmed by analyses of the high-angle annular dark-field scanning transmission electron microscopy (HAADF-STEM) image and the energy dispersive X-ray spectroscope (EDS) line profile [54,55]. In addition, the XRD patterns of AZO and Pt-AZO NPs revealed the crystal structure of a hexagonal wurtzite phase without any secondary or impurity phases [55]. The diffraction peaks of the face centered cubic structure of the Pt crystals were observed for Pt-AZO NPs [55]. Figure 1c shows a TEM image of strip-shaped N-doped graphene with a diameter of ~10 nm. The basal planes were discontinuous and distorted, and some parts were wavy and turbostratic, indicating the presence of defects, which may have facilitated gas diffusion. Cross-sectional SEM images of the Au-WO_3_ and Pt-WO_3_ thin films are shown in Figure 1d,e, respectively. The thickness of the WO_3_ thin film was ~200 nm. The thicknesses of the Pt and Au activator layers deposited on the WO_3_ thin film were estimated to be ~2 nm.

Figure 2 shows the response patterns of the sensing materials exposed to 10 ppm of NO, NO_2_, NH_3_, TMA, and TEA gases. Here, the sensing response is defined as (R_a_ − R_g_)/R_g_, depending on whether the gas is reducing or oxidizing, where R_g_ and R_a_ represent the resistances of the five types of sensing materials in the N-containing compound gases and air, respectively. In this figure, the upward and downward directions of the graph correspond to the decrease and increase of the resistance, respectively. As shown in Figure 2, the responses of the metal oxides (AZO, Pt-AZO, Pt-WO_3_, Au-WO_3_) became positive when they were exposed to reducing gases (NH_3_, TMA, TEA) and negative under exposure to oxidizing gases (NO, NO_2_). This is because all the metal oxides tested in this experiment were n-type semiconductors. Positive and negative responses correspond to the decrease and increase, respectively, of the resistance of the sensing material in the target gas compared with that in air. In contrast, the responses of the N-doped graphene became positive when it was exposed to oxidizing gases (NO, NO_2_), indicating that N-doped graphene is a p-type semiconductor. 

Figure 3, Figure 4 and Figure 5 show graphical representations of the sensing responses (in Figure 2) of the sensors to the N-containing compound gases. Figure 3a presents the sensing responses of AZO NPs to the five N-containing compound gases. The AZO NPs exhibited a decrease in resistance when exposed to 10 ppm TEA, TMA, and NH_3_. Among these, the highest response level was 144 for TEA. The responses were 44 and 24 for TMA and NH_3_, respectively (see Figure 2a). When the AZO NPs were exposed to 10 ppm NO and NO_2_, an increase in resistance was observed, with response values of −0.06 and −0.07 in NO and NO_2_, respectively. 

Figure 3b presents the sensing responses of Pt-AZO NPs to the N-containing compound gases. The Pt-AZO NPs exhibited a higher overall response to the N-based hazardous gases than the AZO NPs (see Figure 2b). Additionally, the Pt-AZO NPs showed a reduced resistance when exposed to 10 ppm TEA, TMA, and NH_3_, with response values of 159, 73, and 23, respectively. When exposed to NO and NO_2_, the resistance increased, and the response was −2.8 and −4.7, respectively. Compared with pure AZO, the Pt-ZNO NPs exhibited almost no change in their response to NH_3_, whereas their response was increased slightly and significantly for the tertiary amines and nitric oxides, respectively. The sensing response of sensing materials can be augmented via noble-metal loading [58]. Consequently, in various gas sensing applications, Pt is loaded as a catalytic additive for enhancing the sensing response [55,59]. In our case, the Pt loading was effective for improving the nitric-oxide sensing. 

Figure 4a presents the sensing responses of the Pt-WO_3_ thin film to the N-containing compound gases. The Pt-WO_3_ exhibited high sensitivity to not only tertiary amines but also nitric oxides. The sensing responses of the Pt-WO_3_ thin film to TEA, TMA, and NH_3_ were 13,277, 3100, and 2489, respectively. The sensing responses to NO and NO_2_ were −481 and −2638, respectively, indicating an increased resistance. Remarkably, the TEA sensing response exceeded ~13,000. To our knowledge, all the responses of the Pt-WO_3_ thin film to TEA, TMA, NH_3_, NO, and NO_2_ are significantly higher than those of previously reported sensing materials based on semiconducting metal oxides (See Table 3).

Figure 4b shows the sensing responses of the Au-WO_3_ thin film to the N-containing compound gases. The sensing responses of the Au-WO_3_ thin film to TEA, TMA, and NH_3_ were 93,666, 9810, and 4821, respectively. The sensing responses of NO and NO_2_ were −0.29 and −0.71, respectively. The Au-WO_3_ thin film exhibited much higher response to TEA, TMA, and NH_3_ than the Pt-WO_3_ thin film. In particular, the Au-WO_3_ thin film exhibited an extremely high sensing response (~100,000) to TEA compared to the other gases. To our knowledge, this is the first report of the highest sensing response to TEA, TMA, and NH_3_, compared to those reported so far, for metal-oxide sensing materials (See Table 3).

Figure 5 shows the sensing responses of N-doped graphene to the N-containing compound gases. NH_3_ and the tertiary amines were not detected, even at a relatively high concentration (10 ppm); only nitric oxides were detected, with a low response of 0.1–0.7. Pure graphene is a p-type semiconductor in air, and exposure to oxidizing gases, such as NO_2_ and O_2_, reduces its resistance by enhancing the hole conduction [60]. Although Lu et al. reported that highly N-doped graphene exhibits n-type semiconducting behavior [61], the sensing response of our N-doped graphene indicated that the sample was a p-type semiconductor. If the doped N atoms replace the C atoms in the hexagonal ring of graphene (quaternary N) efficiently, 2 wt% N in graphene is sufficient to make the material an n-type semiconductor. Thus, our results indicate that the direct substitutional doping was not efficient enough to make the material n-type. When N atoms are doped into graphene, three bonding configurations occur within the C lattice: Quaternary N (direct substitution), pyridinic N, and pyrrolic N [57]. Only quaternary N yields n-type doping; the other two configurations promote p-type doping [62]. 

The XPS N 1s spectra of the N-doped graphene used in our experiments presented that the amount of pyridinic and pyrrolic N was larger than that of quaternary N [57]. As shown in Figure 5, the N-doped graphene exhibited good sensitivity to NO_2_. This was expected, as Shaik et al. reported NO_2_ sensing with N-doped graphene, which was fabricated using a wet process and exhibited p-type behavior [63]. In contrast, the theoretical studies of Jappor et al. and Dai et al. [64,65], which focused on quaternary N-doping, indicated that NO_2_ was weakly physiosorbed onto the N-doped graphene surface. Clearly, the pyridinic and pyrrolic N-doping made graphene a good NO_2_ gas sensor. 

Figure 6 shows the response of AZO NPs, Pt-AZO NPs, Au-WO_3_ thin film, and Pt-WO_3_ thin film at various concentrations of the five N-containing gases (0.1, 1, and 10 ppm) at 400 °C. The sensing response increases with increasing gas concentration. As shown in Figure 6, the ZnO and WO_3_ samples have a lower limit of 0.01 ppm to detect those N-containing gases.

A comparison of the responses of the five sensing materials to 10 ppm of the five N-containing compound gases is shown in Figure 7a–e in the form of radar plots. The radar plots of the sensing response show different patterns for the reducing gases (TEA, TMA, NH_3_) and the oxidizing gases (NO, NO_2_). The specific patterns of the radar plots for the sensing response represent several noteworthy features: (i) The sensing response of the WO_3_ film-based sensor was superior to that of the AZO NP-based sensor for all five N-containing compound gases; (ii) the WO_3_ film and AZO NP-based sensors are more sensitive in detecting TEA compared to the other gases; (iii) the Au-WO_3_ thin film exhibited the highest response for the detection of 10 ppm of TEA, TMA, and NH_3_; (iv) the Pt-WO_3_ thin film showed the best sensing performance for the detection of 10 ppm of NO and NO_2_.

In particular, the sensing response of the WO_3_ film and AZO NP-based sensors increases in the order of NH_3_, TMA, and TEA. The response is significantly higher in detecting TEA compared to the other gases. This can be attributed to an electron donating effect [25]. When the metal oxide sensor is exposed to the reducing gas, the reducing gas reacts with the adsorbed oxygen ions and the free electrons are released back to the conduction band of the metal oxides. This leads to an increase in conductance and consequently an increase in response. At the working temperature of WO_3_ film and AZO NP (400 °C), the O^2−^ ion species mainly interact with the gas molecules [55], according to the following equations for TEA (Equation (1) [24]), TMA (Equation (2) [28]), and NH_3_ (Equation (3) [38]): (1)$$2{\left(C_{2}H_{5}\right)}_{3}N_{\left(\mathrm{ads}\right)}+43O_{\left(\mathrm{ads}\right)}{}^{2-}\to \;12{\mathrm{CO}}_{2}{}_{\left(\mathrm{gas}\right)}+15H_{2}O_{\left(\mathrm{gas}\right)}\;+\;2{\mathrm{NO}}_{2}{}_{\left(\mathrm{gas}\right)}+86e^{-}$$
(2)$$\;2{\left({\mathrm{CH}}_{3}\right)}_{3}N_{\left(\mathrm{ads}\right)}+21O_{\left(\mathrm{ads}\right)}{}^{2-}\to \;6{\mathrm{CO}}_{2}{}_{\left(\mathrm{gas}\right)}+9H_{2}O_{\left(\mathrm{gas}\right)}\;+{\;N}_{2}{}_{\left(\mathrm{gas}\right)}+42e^{-}\;$$
(3)$$\;2{\mathrm{NH}}_{3}{}_{\left(\mathrm{ads}\right)}+3O_{\left(\mathrm{ads}\right)}{}^{2-}\to \;9H_{2}O_{\left(\mathrm{gas}\right)}\;+{\;N}_{2}{}_{\left(\mathrm{gas}\right)}+6e^{-}\;.$$

As a consequence, a number of the released electrons increases in the order of NH_3_, TMA, and TEA. Therefore, the significantly enhanced sensing response to TEA is mainly attributed to the great number of released electrons. 

In addition, the responses of the Au-WO_3_ thin film for sensing TEA, TMA, and NH_3_ are remarkably better than those of the Pt-WO_3_ thin film. To understand this result, we investigated the surface morphology and compositional distribution of those WO_3_ thin films by using FE-SEM equipped with an energy-dispersive X-ray spectroscope (EDS). Figure 8 shows the top-view SEM images of the as-prepared Pt- and Au-WO_3_ thin films. The images show the Pt particles cover the surface of the WO_3_ thin film (Figure 8a), while the Au islands are randomly distributed on its surface (Figure 8b).

Figure 9 and Figure 10 show the elemental distribution at the cross-sectional areas of the Pt- and Au-WO_3_ thin films, respectively. The EDS elemental color mapping results present that the Pt elements cover the entire surface of the film, but the Au elements are sparsely distributed compared to Pt. According to the sensing mechanism, the sensing response of the n-type metal oxide gas sensor mainly depends upon the concentration of oxygen ion species (O^−^ or O^2−^) adsorbed on the surface of the sensing materials. Further, the loaded noble metals provide more active sites for the adsorption of oxygen ion species owing to a spill-over effect. Therefore, too many Pt atoms covered on the film decreases the number of active sites available on the film’s surface, leading to the reduced response. Consequently, the Au-WO_3_ thin film exhibits better sensing response compared to the Pt-WO_3_ thin film for the reducing gases of TEA, TMA, and NH_3_. As a result, we can find that a moderate amount of metal catalyst plays an important role in improving the sensing response. 

More importantly, these sensing responses of the Au-WO_3_ thin film to TEA, TMA, and NH_3_, and the Pt-WO_3_ thin film to NO and NO_2_ are much higher than those of previously reported sensors based on metal-oxide sensing materials (See Table 3). The sensitivities of most metal-oxide sensors reported for the detection of TEA are very low. In addition, there are few reports on TEA detection using WO_3_ materials. For example, polyaniline-WO_3_ nanocomposites exhibited a sensing response of 81 to 100 ppm TEA at room temperature [23]. In the case of TMA sensing, there are many reports showing good response results. Cho et al. reported a high response to 5 ppm TMA: 56.9 at 450 °C for a WO_3_ hollow sphere [29] and 373.74 at 300 °C for MoO_3_ nanoplates [30]. Sensing N-containing compound gases, such as NH_3_ and NO_x_, using Pt-WO_3_ and Au-WO_3_ has been reported. For example, D’Arienzo et al. reported that the sensing response of Pt-WO_3_ to 74 ppm NH_3_ was 110 at 225 °C [34]. Srivastava and Jain found that the sensing response of Pt-WO_3_ to 4000 ppm NH_3_ was 12 at 450 °C [35]. Maekawa et al. reported that Au-WO_3_ (0.8–1.0 wt% Au) exhibited a good sensing response (40–60) to 50 ppm NH_3_ gas at 450 °C [36]. For NO and NO_2_ gas sensing, Penza et al. reported Au-WO_3_ sensing responses of −100.3 (440 ppm) and −6.5 (10 ppm), respectively, at 150 °C [43]. Xia et al. reported that Au-WO_3_ (1 wt% Au) exhibited a high sensing response (approximately −400) to 10 ppm NO_2_ gas at 150 °C [48]. 

Additionally, we evaluated the sensing responses of the five sensing materials to 10 ppm of the five N-containing compound gases, as presented in the form of the radar plots in Figure 11. The response time is defined as the time required to reach 90% of the saturation resistance upon the exposure to full-scale concentration of the gas. As shown in Figure 11, the Pt-WO_3_ and Au-WO_3_ thin films exhibited a fast response time (i.e., a very rapid reaction rate) for the detection of TEA, TMA, and NH_3_. In particular, the Pt-WO_3_ thin film showed high responses to all the N-containing gases, as well as the fastest response (<20 s).

The higher and faster sensing response of the Pt-WO_3_ and Au-WO_3_ thin films is attributed to the addition of an appropriate amount of metal additives to WO_3_, which promoted chemical reactions by reducing the activation energy between the film surface and the target gas. Cu-loaded WO_3_ and Ag-loaded WO_3_ have also been reported to detect N-containing compound gases with high sensitivity [27,42]. Furthermore, the outstanding sensing responses of the Pt-WO_3_ and Au-WO_3_ thin films are attributed to the deposition of high-quality thin films via the dual ion beam sputtering technique. The thin films deposited using this technique exhibited an exact stoichiometry. Therefore, the dense thin-film formation allowed the deposition of high-quality films with a very small thickness. Furthermore, the five plot patterns are significantly different, indicating that the five sensor materials can be used for an e-nose to distinguish the five N-containing compound gases.

## 4. Conclusions

We investigated the sensing properties of five types of sensing materials (AZO NPs, Pt-AZO NPs, a Pt-WO_3_ thin film, a Au-WO3 thin film, and N-doped graphene) for the detection of five hazardous N-containing compound gases (TEA, TMA, NH_3_, NO, and NO_2_). Owing to the different reactivities of the gases, the sensing materials exhibited different sensing response patterns. The metal-oxide sensors of AZO, Pt-AZO, Pt-WO_3_, and Au-WO_3_ showed positive responses to NH_3_, TMA, and TEA (reducing gases) and negative responses to NO and NO_2_ (oxidizing gases). This is because all the metal oxides tested in the experiment were n-type semiconductors. In contrast, the N-doped graphene exhibited a positive response to NO and NO_2_ owing to its p-type semiconducting property. The metal oxide-based materials showed significantly higher sensing responses to the tertiary amines than to the nitric oxides. The N-doped graphene reacted only to the nitric oxides. Among the sensing materials, the Au-WO_3_ and Pt-WO_3_ thin films exhibited the best sensing response. More importantly, the sensing responses of the Au-WO_3_ thin film to TEA, TMA, and NH_3_ and the Pt-WO_3_ thin film to NO and NO_2_ were much higher than those of previously reported sensors based on metal-oxide sensing materials. In particular, the Au-WO_3_ and Pt-WO_3_ thin films exhibited extremely high sensing responses of approximately 100,000 for 10 ppm of TEA and approximately −2700 for 10 ppm of NO_2_, respectively. Accordingly, our study indicates that the five N-containing compound gases can be distinctively detected using the five sensor elements via application of recognition technology that shows different patterns of the sensing response. In order to demonstrate the analytical applicability of the proposed method in a real application, future studies will be conducted to investigate whether the sensor array consisting of the five sensing materials selectively detects only one target gas when mixed with the five N-containing compound gases. In addition, the reproducibility, long-term stability, and humidity interference of the sensor array will be tested. 

## Figures and Tables

**Figure 1 sensors-19-03565-f001:**
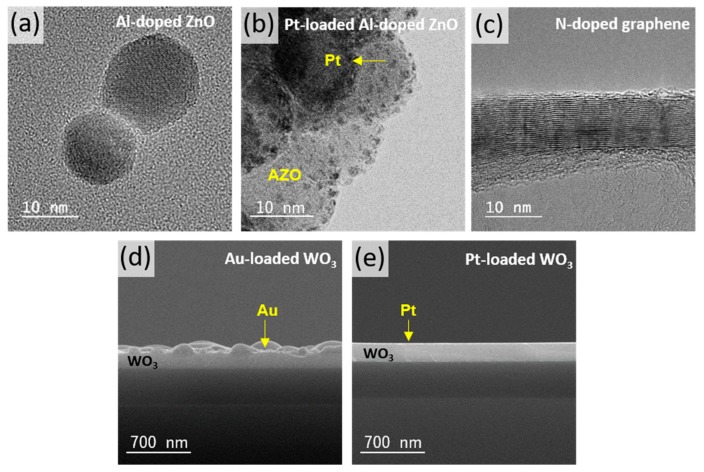
TEM images of (**a**) Al-doped ZnO (AZO) nanoparticles (NPs), (**b**) Pt-AZO NPs, and (**c**) N-doped graphene and SEM images of (**d**) the Au-loaded WO_3_ (Au-WO_3_) thin film and (**e**) the Pt-loaded WO_3_ (Pt-WO_3_) thin film.

**Figure 2 sensors-19-03565-f002:**
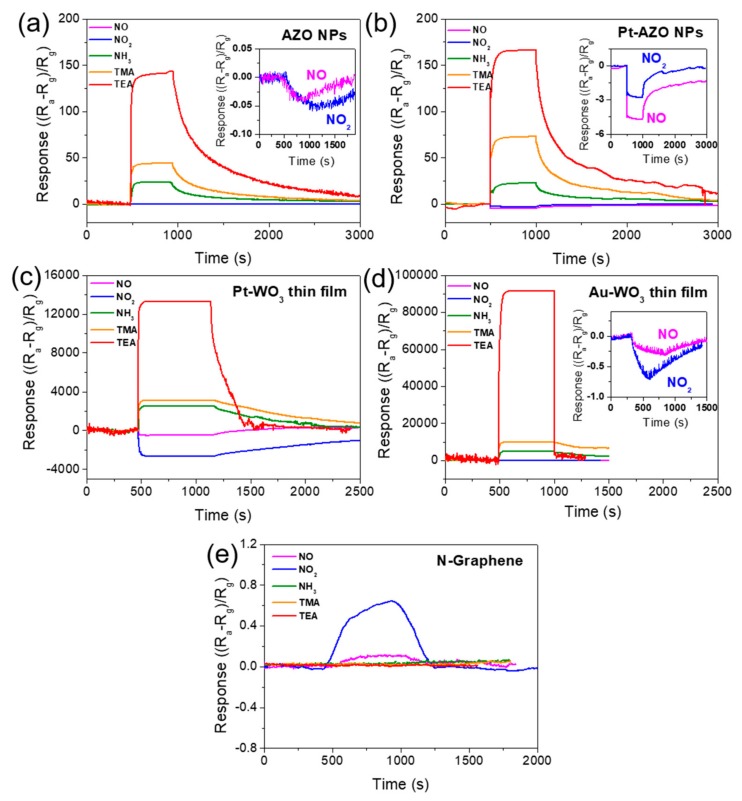
Variation in the sensing responses to 10 ppm of the five N-containing compound gases for the (**a**) AZO NPs, (**b**) Pt-AZO NPs, (**c**) Pt-WO_3_ thin film, (**d**) Au-WO_3_ thin film, and (**e**) N-doped graphene at 400 °C.

**Figure 3 sensors-19-03565-f003:**
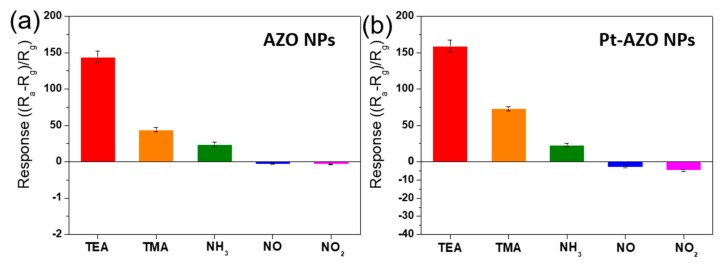
Bar graph of the sensing responses of the (**a**) AZO and (**b**) Pt-AZO NPs to 10 ppm of the N-containing compound gases at 400 °C.

**Figure 4 sensors-19-03565-f004:**
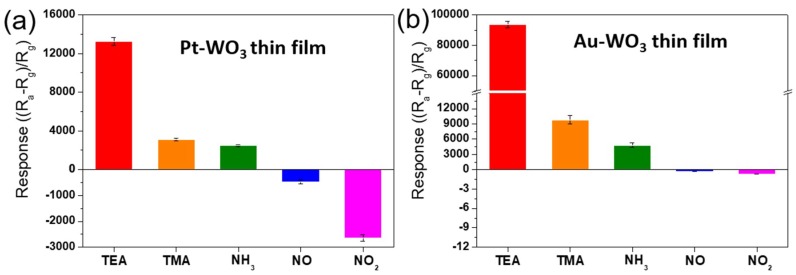
Bar graph of the sensing responses of the (**a**) Pt- and (**b**) Au-WO_3_ thin films to 10 ppm of the N-containing compound gases at 400 °C.

**Figure 5 sensors-19-03565-f005:**
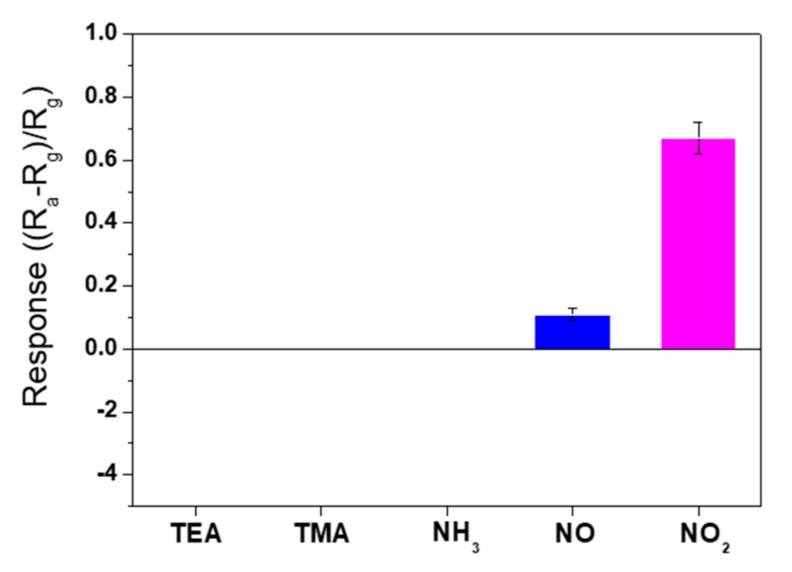
Bar graph of the sensing responses of the N-doped graphene to 10 ppm of the N-containing compound gases at 400 °C.

**Figure 6 sensors-19-03565-f006:**
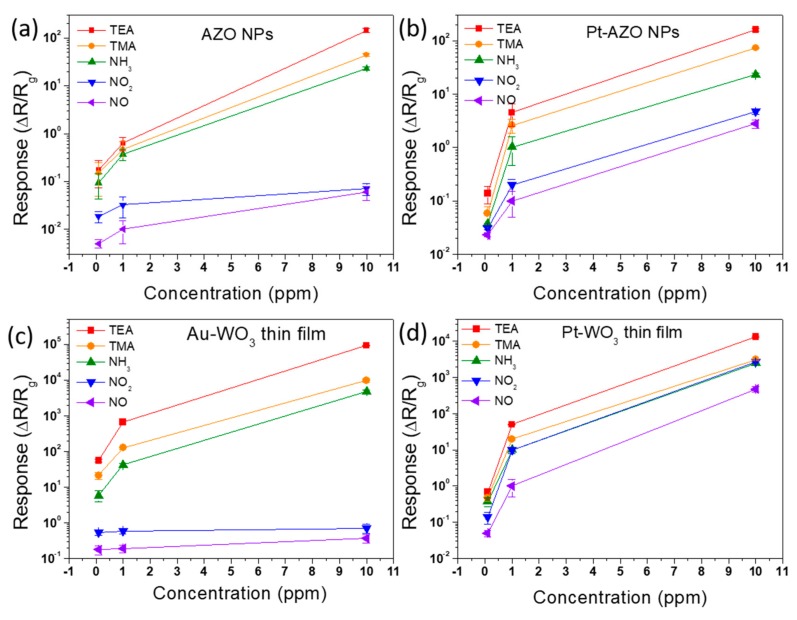
Sensing response of (**a**) AZO NPs, (**b**) Pt-AZO NPs, (**c**) Au-WO_3_ thin film, and (**d**) Pt-WO_3_ thin film to different concentrations of the five N-containing gases at 400 °C.

**Figure 7 sensors-19-03565-f007:**
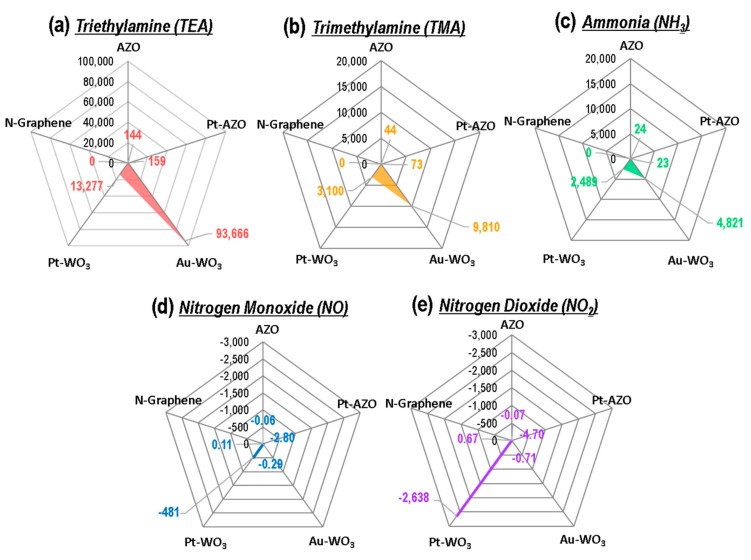
Radar plots of the sensing responses ((R_a_ − R_g_)/R_g_) of the five sensing materials to 10 ppm of (**a**) TEA, (**b**) TMA, (**c**) NH_3_, (**d**) NO, and (**e**) NO_2_.

**Figure 8 sensors-19-03565-f008:**
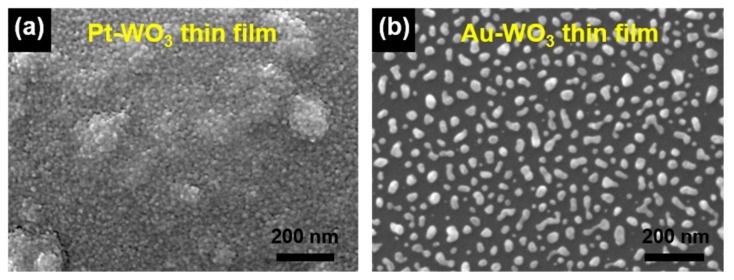
Top-view SEM images of the (**a**) Pt- and (**b**) Au-WO_3_ thin films.

**Figure 9 sensors-19-03565-f009:**
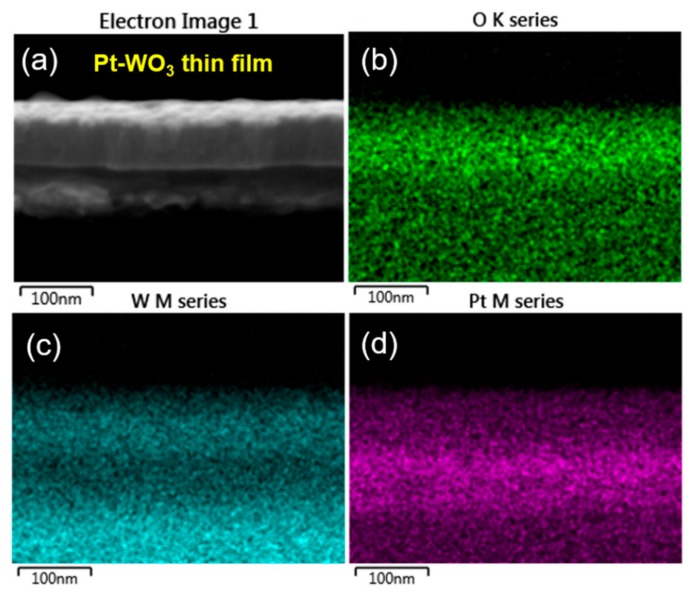
(**a**) Cross-sectional SEM image depicting analyzed region of Pt-WO_3_ thin film and energy-dispersive X-ray spectroscope (EDS) elemental color mapping images for (**b**) O, (**c**) W, and (**d**) Pt elements.

**Figure 10 sensors-19-03565-f010:**
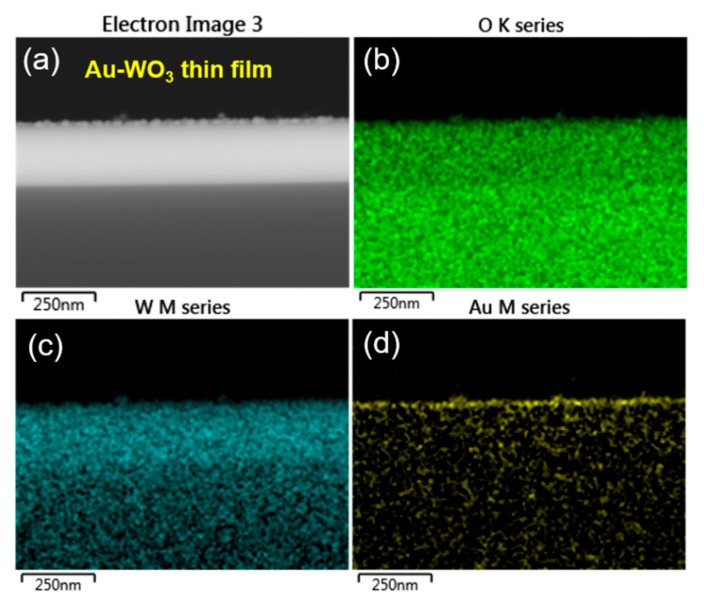
(**a**) Cross-sectional SEM image depicting analyzed region of Au-WO_3_ thin film and EDS elemental color mapping images for (**b**) O, (**c**) W, and (**d**) Au elements.

**Figure 11 sensors-19-03565-f011:**
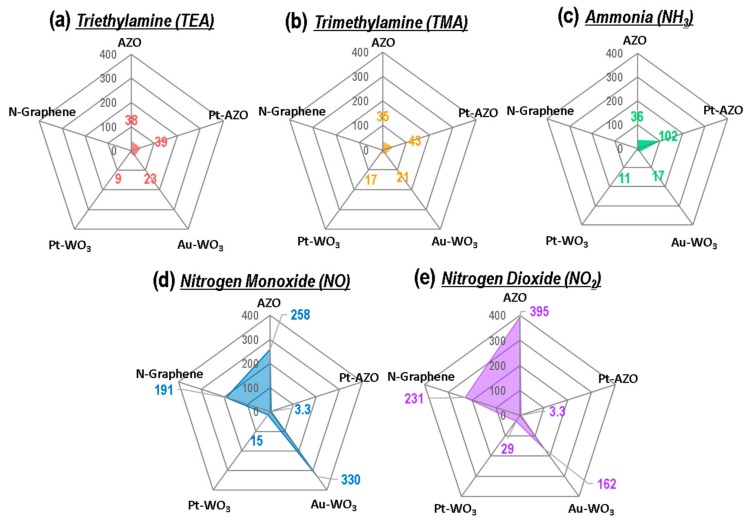
Radar plots of the response time (unit: sec) of the five sensing materials to 10 ppm of (**a**) TEA, (**b**) TMA, (**c**) NH_3_, (**d**) NO, and (**e**) NO_2_.

**Table 1 sensors-19-03565-t001:** Exposure limits for the gases established by the U.S. National Institute for Occupational Safety and Health.

Gas	Short-Term Exposure Limit (15 min, ppm)	Time Weighted Average (8 h, ppm)
NH_3_	50	35
TMA	15	10
TEA	25	15
NO	-	25
NO_2_	5	3

**Table 2 sensors-19-03565-t002:** Fabrication methods and specifications for the five sensing materials.

Sensing Materials	Fabrication Method	Specifications
AZO NPs	Hydrothermal synthesis	AZO NPs Size: 20–30 nm
Pt-AZO NPs	Hydrothermal synthesis + sputtering	AZO NPs size: 20–30 nm
		Pt NPs size: 2–3 nm
		Al doping: 1 at%
Pt-WO_3_ thin film	Dual ion beam sputtering	WO_3_ thickness: 200 nm
		Pt thickness: 2 nm
Au-WO_3_ thin film	Dual ion beam sputtering	WO_3_ thickness: 200 nm
		Au thickness: 2 nm
N-doped graphene	Arc discharge of graphite	N doping: 2 wt%

**Table 3 sensors-19-03565-t003:** Comparison of sensing properties of various types of metal-oxide-based sensors for the detection of the N-containing gaseous compounds (ΔR ≡ (R_a_ − R_g_) or (R_g_ − R_a_), S *: Sensitivity ≡ response/concentration.).

Gas	Sensing Materials	Operating Temperature [°C]	Concentration [ppm]	Response (R/R_g_)	S * [ppm^−1^]	Ref.
**TEA**	Hollow SnO_2_ microfiber	270	100	49.5	0.49	[20]
	ZnO-NiO hetero-nanostructures	250	200	35	0.17	[21]
	Au-Loaded ZnO/SnO_2_ Core-Shell Nanorods	40	50	12.4	0.25	[22]
	polyaniline-WO_3_ nanocomposites	25	100	80	0.8	[23]
	broken In2O3 microtubes	300	100	72.5	0.72	[24]
	CeO_2_-SnO_2_ Nanoflowers	310	200	252.2	1.26	[25]
	Al-doped ZnO (AZO) nanoparticles	400	10	144	14.4	This work
	Pt-loaded AZO nanoparticles	400	10	159	15.9	This work
	Au-loaded WO_3_ thin film	400	10	93,666	9366.6	This work
	Pt-loaded WO_3_ thin film	400	10	13,277	1327.7	This work
**TMA**	TiO_2_	60	400	1.5	0.004	[26]
	membrane nanotubes	290	10	50	5	[27]
	Cu-doped WO_3_ materials	325	5	120	24	[28]
	MoO_3_ nanopapers	450	5	56.9	11.38	[29]
	WO_3_ hollow spheres	300	5	374.74	74.95	[30]
	MoO_3_ nanoplates	330	50	125	2.5	[31]
	SnO_2_-ZnO nanocomposite	208	1000	2552	2.55	[32]
	Al-doped ZnO (AZO) nanoparticles	400	10	44	4.4	This work
	Pt-loaded AZO nanoparticles	400	10	73	7.3	This work
	Au-loaded WO_3_ thin film	400	10	9810	981.0	This work
	Pt-loaded WO_3_ thin film	400	10	3100	310.0	This work
**NH_3_**	WO_3_ Nanoparticles Thinfilm	240	0.5	2.3	4.6	[33]
	Macroporous WO_3_ Thin Films	225	74	110	1.48	[34]
	Pt catalyzed WO_3_ thick films	450	4000	15.5	0.004	[35]
	Au-loaded WO_3_ powder	450	50	39	0.78	[36]
	single-layer MoSe_2_ nanosheet	25	500	1150	2.3	[37]
	SnO Nanoshell	25	200	37.57	0.19	[38]
	SnO_2_ nanostructures	300	800	222	0.28	[39]
	Al-doped ZnO (AZO) nanoparticles	400	10	24	2.4	This work
	Pt-loaded AZO nanoparticles	400	10	23	2.3	This work
	Au-loaded WO_3_ thin film	400	10	4821	482.1	This work
	Pt-loaded WO_3_ thin film	400	10	2489	248.9	This work
**NO**	Ag doped WO_3_	250	40	38.3	0.96	[42]
	Pd doped WO_3_	200	440	100.3	0.23	[43]
	Cu2+/Polyaniline/WO_3_	25	0.04	9.6	240	[44]
	Al-doped ZnO (AZO) nanoparticles	400	10	0.06	0.006	This work
	Pt-loaded AZO nanoparticles	400	10	2.8	0.28	This work
	Au-loaded WO_3_ thin film	400	10	0.29	0.03	This work
	Pt-loaded WO_3_ thin film	400	10	481	48.1	This work
	N-doped graphene	400	10	0.11	0.01	This work
**NO_2_**	WO_3_ thin fim	200	0.01	28	2800	[46]
	plasma-sprayed WO_3_ coating	130	0.45	77	171.1	[47]
	Pd doped WO_3_	200	10	6.51	0.65	[43]
	Au-doped WO_3_ powder	150	10	412	41.2	[48]
	ZnO hierarchical nanostructure	25	20	11.06	0.55	[50]
	SnO_2_ nanoslab	300	10	120	12	[51]
	Al-doped ZnO (AZO) nanoparticles	400	10	0.07	0.007	This work
	Pt-loaded AZO nanoparticles	400	10	4.7	0.47	This work
	Au-loaded WO_3_ thin film	400	10	0.71	0.07	This work
	Pt-loaded WO_3_ thin film	400	10	2638	263.8	This work
	N-doped graphene	400	10	0.67	0.07	This work

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
