# Peer review of "Selective Detection of Nitrogen-Containing Compound Gases"

_sensors, 2019, doi:10.3390/s19163565_

Round 1

Reviewer 1 Report

The manuscript is dealing with different oxide-type systems for sensing nitrogen based toxic gases and in this respects has major importance.

Although, many aspects should be improved, as follows:

--The original aspects are difficult to be assessed.

--The state of the art is not actual and need to be refreshed with recent publications; I suggest a Table with the compounds/systems that were tested against these gases, their best performances and references.

--Section 2.1.2. What are the Pt NPs size and characteristics?

--Please verify the text: "Pt NPs were coated at a deposition rate of 6−7 nm/min" and explain the technique to do this.

--Is the method described in sub-section 2.1.3. original? No reference is given.

--In order to be sound, the TEM image (Figure 1) of each compounds has to be compared with the before doped one.

-- What quantity of N-containing gases are tested as represented in figure 2?

--The following text needs references for what is known: "To the best of our knowledge, this finding is the first to indicate that AZO NPs can detect TEA sensitively and distinguish it from NOx and NH3."

--Is Figure 6 your own? If not, you have to ask for permission.

---Please describe the mechanism for TEA detection.

-- No interference was studied, not even humidity.

Reviewer 2 Report

Authors and co-worker's paper entitled "Selective Detection of Nitrogen-Containing Compound Gases" reported a method of detecting N-containing gaseous compounds such as TMA, TEA, NH3, NO, NO2,  by comparing the sensing responses of five sensor materials including AZO NPs, Pt-AZO NPs, Pt-AZO thin film, Au-WO3 thin film, and N-doped graphene.  In my opinion, the impact of this paper is limit, and it is objective not completely clear because the manuscript lack of detection of the real sample (Air) to provide the selectivity, robustness, practicality, and reliability. Although the research maybe has several disadvantages in selectivity for mixing gases, the manuscript relatively offers a critical issue to concern the importance of detecting the N-containing gaseous compounds. I recommend the publication of this paper after the subsequent revisions are addressed:

Suggestion comment:

1. Usually, to demonstrate the analytical applicability of the proposed method in a real sample is available. If the author could not provide the data instantly, I suggest the author could serve as the future works in the conclusion. 

2. Is the device reusable in the interday or intraday? Additionally, is the standard deviation was obvious in each fabricated sensor?  It means that the system is reliable or not. 

3. In the abstract and conclusion, the authors state "The Au- and Pt-WO3 thin films exhibited extremely high responses of approximately 100,000 to 10 ppm TEA and approximately -2700 to 10 ppm, NO2, respectively."  I think the sensing response signal was not "ppm" (concentration unit).  

4. The authors should provide reasons to illustrate the properties to compare the advantage and disadvantage of sensing materials. For example, why the response intensity by using Au-WO3 for sensing TEA was better than Pt-WO3?  

5. I strongly suggest that the authors could provide all the calibration curve of the N-containing gaseous compounds for each sensor materials, and define the linear range, Linear regression (R),  and limit of detection (LOD).  

6. Is the relative humidity would influence the accuracy and precision of detection? 

7. In Figure 3-5, the author should supplement the standard deviation in each bar.  

8. A challenge problem, the results may be showed the poor selectivity when the mixture of the gaseous compound, how to conquer this pivotal drawback. 

Editing comment:

1. The introduction is too wordy. Recommend to concise the paragraph into four paragraphs.  

2. Line 89, Tabel 1., correct the "H3" to "NH3."

3. Line 132, the subtitle should be corrected to "characterization."

4. Notice the subscript in the chemical symbol for WO3 in the caption of  Figure 4 and 5.  and Line 192, NH3.

Reviewer 3 Report

Yoo et. al provide an interesting manuscript regarding the detection of nitrogen containing gases. The paper is generally of good quality but lacks some important details and consideration that, in my opinion, the authors need to address prior to publication:

1.       Please add details and maybe a SEM image of the sensing structure. Please add details as to the spacing, the structures etc. How do you know the temperature of the sensing layers and how do you control it?

2.       Please also add experimental details regarding the sputtering of WO3 film, including the respective catalysts. Did you perform EDX/XRD to confirm structure of WO3? Since tungsten oxide may appear in various oxidation states, this information is important.

3.       Do the authors mean synthetic air when writing “clean air”?
In any case, the authors will have to perform the gas sensitive characterization taking into account humidity. Since H2O is almost omnipresent, it is paramount to check its influence on the selectivity. Please add at least a measurement including 50% r.H at 20°C, or similar. Also, it would be nice to see other concentration steps as well, such as to estimate the sensitivity of the different layers. In the representation of the layer’s response a log-scale might be beneficial.

4.       The discussion regarding the surface chemistry of the layers is not sufficient and needs to be added. Regarding the selectivity towards NO/NO2 the authors should take into account SNB Volume 229, 57-62 (2016) and compare the results with the ones achieved here. What about using lower temperatures in the gas sensing experiments? Why did the authors limit all results to 400°C?

Round 2

Reviewer 1 Report

The manuscript entitled Selective Detection of Nitrogen-Containing Compound Gases was thoroughly revised or detailed explanations were offered, so that I can agree to be accepted for publishing, in the actual improved form.

Reviewer 2 Report

    Authors and co-worker's paper entitled "Selective Detection of Nitrogen-Containing Compound Gases" reported a method of detecting N-containing gaseous compounds such as TMA, TEA, NH3, NO, NO2, by comparing the sensing responses of five sensor materials including AZO NPs, Pt-AZO NPs, Pt-AZO thin film, Au-WO3 thin film, and N-doped graphene. Reviewing carefully, I recommend the publication of this paper.

Reviewer 3 Report

After the revision I recommend the manuscript for publication as is.